# Current and Novel Therapeutic Approaches for Treatment of Neovascular Age-Related Macular Degeneration

**DOI:** 10.3390/biom12111629

**Published:** 2022-11-03

**Authors:** Reem H. ElSheikh, Muhammad Z. Chauhan, Ahmed B. Sallam

**Affiliations:** 1Department of Ophthalmology, Harvey and Bernice Jones Eye Institute, University of Arkansas for Medical Sciences, Little Rock, AR 72207, USA; 2Department of Ophthalmology, Cairo University Hospitals, Cairo 11261, Egypt

**Keywords:** neovascular age-related macular degeneration, neovascular AMD, wet AMD, anti-VEGF, faricimab

## Abstract

Age-related macular degeneration AMD is one of the leading causes of blindness in the elderly population. An advanced form of AMD known as neovascular AMD (nAMD) is implicated as the main attributor of visual loss among these patients. The hallmark feature of nAMD is the presence of neovascular structures known as choroidal neovascular membranes (CNVs), along with fluid exudation, hemorrhages, and subretinal fibrosis. These pathological changes eventually result in anatomical and visual loss. A type of proangiogenic factor known as vascular endothelial growth factor (VEGF) has been known to mediate the pathological process behind nAMD. Therefore, therapy has transitioned over the years from laser therapy that ablates the lesions to using Anti-VEGF to target the pathology directly. In this work, we provide an overview of current and emerging therapies for the treatment of nAMD. Currently approved Anti-VEGF agents include ranibizumab, aflibercept, and brolucizumab. Bevacizumab, also an Anti-VEGF agent, is used to manage nAMD even though this is an off-label use. While Anti-VEGF agents have provided a favorable prognosis for nAMD, they are associated with a substantial financial burden for patients and the healthcare system, due to their high cost as well as the need for frequent repeat treatments and visits. Emerging therapies and studies aim to extend the intervals between required treatments and introduce new treatment modalities that would improve patients’ compliance and provide superior results.

## 1. Introduction

Age-related macular degeneration (AMD) is the leading cause of legal blindness in North America and Europe, affecting approximately 10% of patients over 65 years [1]. With increasing life expectancy, the socioeconomic burden imposed by the disease on patients and the health care system is expected to rise [2]. Neovascular AMD is classified according to the severity of the clinical picture into A- early AMD, characterized by the presence of pigmentation abnormalities of the retinal pigment epithelium (RPE) or drusen lipid deposits under the retina; B- intermediate AMD, characterized by the presence of large drusen or geographic atrophy of the RPE (not involving the center of the fovea; C- advanced AMD, the most severe variant and is vision-threatening. Advanced AMD comprises neovascular AMD (nAMD) and non-neovascular AMD with geographic atrophy. Neovascular AMD (nAMD) is responsible for 90% of severe vision loss and blindness caused by AMD [3,4].

The pathology of nAMD is characterized by the presence of subretinal exudates, edema, and hemorrhages involving both the intra- and subretinal spaces. Choroidal neovascularization (CNV) is a hallmark of nAMD, with the growth of pathological leaking blood vessels beneath the retina. The pathology of nAMD is multifactorial, with an interplay between age, metabolic, genetic, and environmental factors. Neovascularization, or the process of angiogenesis, is believed to be initiated by inflammation, hypoxia, and other immune reactions. Inflammatory mediators, including mast cells, neutrophils, and macrophages, produce proangiogenic factors, namely vascular endothelial growth factor (VEGF). Thus, current therapies target binding and neutralizing VEGF to treat the pathology. The VEGF family in mammals includes five primary mediators VEGF-A, placenta growth factor (PGF), VEGF-B, VEGF-C, and VEGF-D. Yet, VEGF-A was the first one discovered and was known to be the primary mediator for angiogenesis and thus was the main therapeutic target; PGF also plays a role in pathological angiogenesis and has been the target of recent research. The VEGF family acts on two central receptors VEGFR1 and VEGFR2. VEGF-A binds to VEGFR-2 stimulating angiogenesis, while PGF binds to VEGFR-1, initiating a signaling pathway that also leads to pathological angiogenesis. PGF also attracts other inflammatory mediators, in turn stimulating the VEGFR-1 present on monocytes [5,6]. Recently, evidence has emerged that the continuous blockade of VEGF-A leads to a compensatory increase in other forms of VEGF, such as VEGF-C and VEGF-D, which have also become therapeutic targets [7,8]. While the discovery of Anti-VEGF agents created a shift in the management of nAMD, treatment responses were variable among patients indicating that other pathological factors could be in play. Angiopoietin-2 (Ang2) is another mediator that plays a vital role in the angiogenesis pathway. It is produced by endothelial cells and plays a role in de-stabilizing vessels by inhibiting Tie2 and competing with Ang1. Ang2 also enhances the vascular response to VEGF-A promoting vascular permeability and angiogenesis. It is part of the Angiotensin1 (Ang1)/Tie2 signaling axis [9]. Hypoxia upregulates both VEGF and Ang2, where both mediators are found to be in increased amounts in the eyes of patients with nAMD [10]. Figure 1 summarizes different pathophysiologic mechanisms involved in the development of AMD.

The goal in the management of nAMD is to reduce vision loss while optimizing the vision-related quality of life. The management of nAMD previously pivoted around laser coagulation and photodynamic therapy (PDT). Scientific advances have improved the visual prognosis of the disease. Anti-VEGF agents have completely changed the management paradigms, and through several landmark pivotal clinical trials, they have become the accepted standard of care (Figure 2) [11]. While improving visual prognosis in patients with nAMD anti-VEGF came with a substantial price tag burdening the patient and the healthcare system. It is of note that repeat injections are needed due to the chronic nature of the disease and the half-lifetime of the Anti-VEGF agents used for therapy. However, this may be counterproductive in terms of patients’ compliance and visual outcomes. Continuous injection of anti-VEGF agents may lead to the development of fibrosis and scarring also hindering visual results. Other complications related to frequent anti-VEGF injections include increased intraocular pressure, ocular inflammation, and endophthalmitis [12].

The so-called ‘poor responders,’ is a subgroup of nAMD patients exhibiting suboptimal results despite consistent dosing regimens. These poor responders were hypothesized to have developed anti-VEGF resistance. The rate of development of resistance to Anti-VEGF treatment is variable and may occur following any number of repeat injections. For this cohort of patients, it is important to address alternative treatment targets independent of inhibiting the VEGF pathway to halt the progression of nAMD [13]. 

Continuous advancements have provided several established and newer emerging anti-VEGF agents [14]. The goal of several clinical trials, complete and ongoing, has been to compare the efficacy and safety of varying formulations and concentrations, optimize dosing and follow-up regimens, and balance visual outcomes with the economic restraints of medication cost and physician visits. While most of the controlled clinical trials yielded excellent results, a discrepancy is seen in real-life clinical studies for several reasons, including the absence of strict inclusion/exclusion criteria and less optimum patient compliance compared to the more idealized clinical trial setting [15]. Researchers aim to fine-tune our current treatment protocols to bridge this gap and attain similar results to clinical trials to maximize visual outcomes and decrease the number of visits and treatments [16]. This review highlights the important current and emerging treatment modalities in managing patients with nAMD.

## 2. Laser Therapy

Before the development of anti-VEGF agents, laser photocoagulation was considered an important treatment option for patients with nAMD. The macular photocoagulation study (MPS) was a series of three randomized, controlled clinical trials to evaluate improvement in BCVA in patients with nAMD treated with laser. The Argon study evaluated whether argon blue-green photocoagulation was beneficial in preventing or delaying vision loss in patients with AMD when applied at 200 to 2500 microns from the foveal avascular zone FAZ (extrafoveal). The relative risk of severe vision loss over three years was 1.4 in the non-laser group versus the laser-treated group (95% CI 1.1–1.9) [17]. The Krypton study assessed krypton red photocoagulation in preventing loss of central vision when applied at CNVs with posterior border 1 to 199 microns from the FAZ in patients with AMD (juxtafoveal). Results showed that the relative risk of losing six or more lines from baseline to follow-ups conducted between 6 months-5 years was 1.2 (*p* = 0.04) in the non-laser group versus the laser-treated group. 

The Foveal study determined whether laser photocoagulation was beneficial in preventing or delaying further visual losses in patients with treatment-naive or recurrent CNVs under the center of the FAZ (subfoveal). After 24 months, the laser group and the non-laser group lost three and four lines of VA, respectively (*p* = 0.003). The general principle behind laser therapy was to thermally ablate the neovascular membrane. While theoretically, this addresses the pathology, laser caused collateral damage to the adjacent parts of the macula resulting in visual field damage. Thus, using laser coagulation was not a viable option for sub-foveal lesions, which are more frequent than the extra- and juxta foveal lesions [18].

## 3. Photodynamic Therapy

Photodynamic Therapy (PDT) was approved in 2000 to treat patients with nAMD. It entails using injected porphyrin (commonly verteporfin and porfimer sodium) dyes that later become laser-activated by a specific wavelength to achieve the required clinical effects [19]. Based on FA, three angiographic subtypes of CNV have been described: A-Classic CNV that has classic components >50% of the total lesion of CNV area; B- Minimally classic CNV has <50% of the total lesion area comprising a classic subtype; C-3. Occult with no classic component. The safety and efficacy of PDT in patients with nAMD have been evaluated in several important clinical trials, Table 1. 

Blinder and colleagues conducted a meta-analysis of the results of the TAP and VIP studies to clear the discrepancies in the findings of both studies. The TAP study had previously concluded that PDT is of clinical benefit to predominantly classic CNVs and occult lesions, while minimally classic lesions did not benefit from PDT. However, the VIP study showed that improvement was more evident in eyes with smaller lesion sizes (less than 4-disc areas) and visual acuity (VA) ≤ 20/50. Results of this metanalysis revealed that vision loss with PDT was less with smaller lesions, suggesting that the PDT could be used for smaller lesions irrespective of their angiographic subtype [24].

Most of the studies on PDT in patients with nAMD concluded that although vision loss could be reduced, vision did not necessarily improve, with repeated treatments required to maintain results. Overall, the safety profile of PDT was good without significant side effects. Reported side effects were mainly visual disturbances, photosensitivity reactions (patients are advised to avoid sunlight for 48 h after the infusion), and back pain experienced during the infusion. Necrosis due to extravasation of verteporfin was also a feared side effect of treatment [25].

Later, with the introduction of anti-VEGF agents, the use of PDT was dramatically reduced, being reserved for specific situations, including as an adjunct therapy with other anti-VEGF agents, such as the FOCUS and the ANCHOR study, in patients with a contraindication to the use of intravitreal anti-VEGF agents, and in patients with conditions as polypoidal choroidal vasculopathy (PCV) in which PDT offered promising results. The role of PDT in the management of patients with PCV will be discussed later in the EVEREST and EVEREST 2 trials in later sections in this review, but in general, the use of PDT has been replaced by anti-VEGF agents for the management of patients with nAMD.

## 4. Pegaptanib

Pegaptanib (Macugen; Eyetech, Palm Beach Gardens, FL). Pegaptanib selectively binds and neutralizes VEGF 165. It was the first FDA-approved Anti-VEGF agent for the treatment of nAMD in 2004. The VISION study consisted of two randomized, double-masked, controlled clinical trials to evaluate the efficacy of pegaptanib in patients with nAMD. There were broad inclusion criteria, and patients with all angiographic subtypes of nAMD were included in the study. Patients were randomized to receive pegaptanib (at 0.3 mg, 1.0 mg, and 3.0 mg doses) or sham injections over 48 weeks at six weeks intervals. The study’s primary endpoint was the proportion of patients losing fewer than 15 letters. All three treatment groups had significantly better results than the control group. In the group receiving the 0.3 mg injection, 70% lost less than 15 ETDRS letters compared to 55% in the control group. Reported side effects included anterior chamber inflammation, corneal edema, and decreased VA. Other side effects that were related to intravitreal injections included endophthalmitis (1.3%), retinal detachment (0.7%), and traumatic injury of the lens (0.6%) [26].

While pegaptanib was the first approved anti-VEGF agent for the treatment of nAMD (similar to PDT), results were limited to preventing visual loss with no observable improvement in visual or anatomical results. This led to trying a combination therapy of pegaptanib with PDT in two studies with improvement in VA in 60% of patients, which is superior to when each of those modalities was used separately [27].

## 5. Bevacizumab

Bevacizumab (Avastin, Genentech, South San Francisco, CA, USA) is a full-length humanized monoclonal antibody that the FDA has approved for managing metastatic colon cancer. However, it is used, off-label, for the management of nAMD by injecting it intravitreally [28]. Bevacizumab works by binding and neutralizing the VEGF-A isoform. 

The SANA study assessed the efficacy and safety of systemic bevacizumab treatment for patients with nAMD. Patients were given an initial intravenous infusion of bevacizumab (5 mg/kg) followed by 1–2 doses two weeks apart. VA had improved by the first two weeks. By week 24, the observed increase in early treatment diabetic retinopathy study (ETDRS) letters was +14 letters (*p* < 0.001), and the mean optical coherence tomography (OCT) central retinal thickness (CRT) decreased by 112 μm (*p* < 0.001). The main observed side effect was a rise in systolic and diastolic blood pressure (+11 mmHg, *p* = 0.004; +8 mmHg, *p* < 0.001, respectively) at three weeks, this was easily controlled using antihypertensives. Although the overall results were satisfying, the widescale use of systemic bevacizumab was not applicable for treating nAMD due to possible side effects [29].

The first case report for the off-label intravitreal use of bevacizumab as a treatment for nAMD was published in 2005. A patient with nAMD who was responding poorly to pegaptanib received intravitreal bevacizumab (1.0 mg). OCT and VA were assessed after one week. OCT revealed the resolution of subretinal fluid (SRF), which remained stable for four weeks, along with VA [30]. Spaide and colleagues conducted a retrospective review on patients with nAMD that were treated with intravitreal bevacizumab. VA using Snellen letters and OCT examinations were used for assessment and follow-up. The mean VA at baseline was 20/184; the mean VA was 20/137, 20/122, and 20/109 at the 1, 2- and 3-months follow-up, respectively (*p* < 0.001). At baseline, the mean central macular thickness was 340 μm and decreased to a mean of 213 μm by the third month (*p* < 0.001). Although these results did not reflect long-term outcomes, they did show favorable short-term visual and anatomic outcomes in the management of patients with nAMD using bevacizumab [31]. 

The IVAN study was a European multi-center randomized non-inferiority trial comparing the efficacy of bevacizumab to ranibizumab. Patients with nAMD were randomized to receive either intravitreal ranibizumab or bevacizumab. Patients were also randomized to receive either a monthly dosing regimen or a pro re nata (PRN) regimen. Regarding best-corrected visual acuity (BCVA), bevacizumab was non-inferior nor inferior to ranibizumab (mean difference −1.37 letters, *p* = 0.26). The PRN dosing was also non-inferior nor inferior to the continuous dosing regimen (−1.63 letters, *p* = 0.18). The frequency of arterial thrombotic events and the overall safety profile were similar in both groups [32].

The CATT study was a randomized, multi-center clinical trial evaluating the efficacy of ranibizumab and bevacizumab when administered monthly or PRN in patients with nAMD. It also evaluated the effect of switching to a PRN dosing regimen after one year of monthly dosing. Patients were randomized to receive either bevacizumab or ranibizumab and a monthly or PRN dosing regimen. At one year, patients on the monthly regimen were randomized to either a monthly regimen or a PRN regimen. Among patients following a similar dosing regimen for the two years, there was no statistically significant difference between the two drugs (bevacizumab-ranibizumab difference, −1.4 letters; *p* = 0.21). The mean gain in VA was greater in the monthly dosing regimen compared to PRN (difference, −2.4 letters *p* = 0.046). While switching from monthly to as-needed treatment resulted in a greater mean decrease in vision during year two (−2.2 letters; *p* = 0.03) and a lower proportion without fluid (−19%; *p* < 0.0001). The death and thrombotic events rates were similar in both drug groups (*p* > 0.60). The proportion of patients with one or more serious systemic adverse events was higher with bevacizumab when compared to ranibizumab, 39.9% vs. 31.7% (*p* = 0.009). However, most of these adverse events were not previously reported as linked to systemic anti-VEGF therapy. The CATT study demonstrated that over a 2-year follow-up period the two drugs were similar in terms of VA, and the rates of death, stroke, and myocardial infarction did not differ between the two groups [33].

The BRAMD study was a randomized, multicenter, controlled, double-masked clinical trial comparing the efficacy of ranibizumab to bevacizumab in the treatment of nAMD. The patients were randomized to receive either monthly ranibizumab or bevacizumab injections for one year. The mean gain was 5.1 (±14.1) letters in the bevacizumab group and 6.4 (±12.2) letters in the ranibizumab group (*p* = 0.37). There was no statistically significant difference in either absolute CRT or CRT change (*p* = 0.13), presence of subretinal (*p* = 0.14), or intraretinal fluid (*p* = 0.10). Adverse events were similar in both groups [34].

The LUCAS treat-and-extend (TREX) protocol evaluated and compared the efficacy of ranibizumab versus bevacizumab when administered in a treat-and-extend protocol in patients with nAMD. This was a multicenter randomized inferiority trial; the inferiority limit was set at 5 ETDRS letters. Patients were given monthly injections of either ranibizumab or bevacizumab until the disease was inactive. Afterward, treatment was extended gradually by two weeks intervals at a time with a maximum of 12 weeks. Any recurrence decreased the interval period by two weeks. Recurrent disease was defined as any fluid on OCT, new or persistent hemorrhage or dye leakage, or increased lesion size on fluorescein angiography (FA). Deterioration of BCVA was not considered a criterion for recurrence. Bevacizumab and ranibizumab were equal in BCVA with 7.9 and 8.2 letters gained, respectively (*p* = 0.845). The difference in measured CRT was not statistically significant between the two drugs: −112 μm for bevacizumab and −120 μm for ranibizumab (*p* = 0.460). The number of treatments required was 8.9 for bevacizumab and 8.0 for ranibizumab (*p* = 0.001). Thrombotic events were fewer in the ranibizumab (1.4%) group than in the bevacizumab group (4.5%) (*p* = 0.05), while the ranibizumab group had significantly more cardiac events (*p* = 0.036). Thus, bevacizumab and ranibizumab had similar efficacy using a treat-and-extend protocol. This is promising as monthly visits for other regimens are taxing to patients and the healthcare system [35].

The previous studies showed that bevacizumab was non-inferior to ranibizumab. Thus, it could be considered an effective and non-expensive treatment for nAMD. However, its use for this purpose remains off-label and variable in different countries according to regulatory measures. A cost-effectiveness study conducted in Europe compared aflibercept, ranibizumab, and bevacizumab in the treatment of nAMD concluding that the bevacizumab was the most cost-effective modality. The cost for aflibercept was €943 per injection. However, €533 was the highest price for aflibercept to make it cost-effective. It has been estimated that in choosing aflibercept over bevacizumab for the treatment of nAMD, Europe overspends €335 million annually [36].

## 6. Ranibizumab

Ranibizumab is a humanized, recombinant fragment of a monoclonal antibody (Fab) with an affinity to VEGF. Ranibizumab neutralizes all isoforms of VEGF-A (including the VEGF soluble fragments 110,121, and 165 as well as tissue-bound isoforms) with its specific binding at amino acid sites 88 and 89. After intravitreal injections, ranibizumab was found to penetrate the retina more efficiently and reach the subretinal space than the larger whole antibody [37]. Ranibizumab is also considered safe owing to its short half lifetime (2–4 days) as opposed to roughly three weeks for bevacizumab and its rapid systemic clearance [37]. Ranibizumab, unlike the whole antibody, does not contain the Fc (fragment crystallizable portion) in its structure, which is responsible for binding complement [38]. Thus, preventing complement-associated intraocular inflammation after intravitreal injections.

Several studies have evaluated the outcomes of intravitreal ranibizumab in the management of neovascular AMD. The MARINA study [39] was a randomized, double-blinded, controlled, multicenter phase 3 clinical trial aimed to investigate the response of patients with nAMD with minimally classic or occult CNV (with no classic lesions) to ranibizumab injections. A total of 716 patients were randomized to receive either 0.3 mg ranibizumab, 0.5 mg ranibizumab, or sham injections monthly over two years. At 24 months, 33.3% of patients receiving 0.5 mg and 26.1% receiving 0.3 mg gained at least 15 ETDRS letters compared to only 3.8% in the sham group (*p* < 0.001 for all comparisons). At the same time, 90% of patients treated with ranibizumab had lost fewer than 15 letters compared to the 53% in the sham group (*p* < 0.001). The VA improvement was in minimally invasive and occult CNV, independent of membrane type, membrane size, or the initial baseline VA. While ranibizumab was not seen to cause regression of neovascularization, it has been shown to stop the growth of CNV [39].

The multicentric randomized ANCHOR trial evaluated the efficacy of intravitreal ranibizumab in patients with predominantly classic CNVs [40]. A total of 432 patients were randomized to receive either PTD with verteporfin every three months as needed plus a monthly sham injection, or sham PTD as needed every three months and a monthly injection of (0.3 mg ranibizumab, 0.5 mg ranibizumab). The study essentially compared monthly injections of both doses of ranibizumab to standard PDT every three months (if leakage was seen by angiography). At 24 months, 90% of patients treated with ranibizumab had lost < 15 ETDRS letters versus 65% in the PTD group (*p* < 0.001). Approximately 41% of patients in the 0.5 mg ranibizumab group versus 36% in the 0.3 mg ranibizumab group versus 6% in the patients treated with PTD gained at least 15 letters (*p* < 0.001 for all comparisons). VA at baseline and the size of the lesions were not determinants of the outcome [40]. The HORIZON study was an open-label extension trial offered to patients who completed the MARINA and the ANCHOR studies. In this study, the patients were provided monthly injections as needed. Preliminary results showed that patients generally required injections within the first six months. 

The rationale behind the injection of anti-VEGF is to inhibit the progression of the lesions. Still, since it’s not essentially a curative approach to the root pathology, it’s not expected that patients would reach a point where they can stop injections indefinitely. However, monthly injections for life are not a reasonable or financially feasible option to maintain visual acuity and prevent the progression of the pathology. This led to a series of studies attempting a less frequent dosing regimen while maintaining visual outcomes. The PIER study investigated the option of a less frequent dosing regimen. Patients with sub-foveal CNV were randomized to receive either 0.3 mg ranibizumab, 0.5 mg ranibizumab, or sham intravitreal injection. Participants received three ranibizumab/sham intravitreal injections every four weeks, followed by injections every three months. At the 12-month follow-up, there was a mean reduction of 1.6 and 0.2 letters (in the 0.3 and 0.5 mg groups, respectively). The ranibizumab-treated patients retained better vision than those receiving sham injections, who lost 16.3 letters on average during the same follow-up period (*p* < 0.0001 for all comparisons). Overall results of this trial were worse than the ANCHOR and the MARINA studies when patients received monthly injections. Thus, the results of the PIER study suggested that dosing injections every three months was not a suitable approach. Another similar smaller trial, the EXCITE trial, aimed to prove the noninferiority of quarterly ranibizumab injections as opposed to monthly injections in patients with sub-foveal CNVs secondary to nAMD. After 12 months of follow-up, the trial failed to prove the non-inferiority of the quarterly regimen [41]. In search of a better-optimized strategy for spaced re-injections, the PRONTO study [42] included 40 patients with sub-foveal CNV to receive three monthly injections of ranibizumab 0.5 mg. OCTs were performed at baseline and at least monthly after injection. FAs were obtained at baseline and every three months thereafter. Afterward, re-injections would be undertaken only if the patient had one or more of the following: increase of CRT of ≥100 μm, loss of at least five letters of visual acuity, the persistence of sub- or intraretinal fluid one month after the last injection, new hemorrhage in the macula, or new onset classic CNV. At 24 months, the visual acuity had significantly improved by a mean of 10.1 ETDRS letters (*p* < 0.001). The PRONTO study’s visual outcomes were comparable to the ANCHOR and MARINA studies. The average annual injections in the PRONTO study was 5.6 versus 13 in the MARINA study.

Several trials aimed to further evaluate the PRN dosing regimen and if it can be applied in treating patients with AMD to reduce the financial burden and frequency of visits without compromising the results. The SAILOR study was a multicenter, randomized (for one cohort), and open-label for a second cohort. The first cohort was randomized to receive either 0.3 mg or 0.5 mg ranibizumab for three monthly loading doses; further treatment was based on VA and OCT findings. The second cohort received three monthly doses of 0.5 mg ranibizumab, and further treatment was based on physician discretion [43]. In each cohort, patients were divided according to previous treatment status (treatment naïve and previously treated). In cohort 1 at 12 months, 14.6% of the 0.3 mg group and 19.3% in the 0.5 mg group had gained >15 ETDRS letters (treatment naïve). In the previously treated group, 15.8% in the 0.3 mg group and 16.5% of the 0.5 mg group had gained >15 ETDRS letters. Overall, the results of the SAILOR study suggested that PRN dosing results were inferior to previous trials with a fixed dosing regimen. In the SUSTAIN trial, three monthly injections of ranibizumab (0.3 mg, which was later switched to 0.5 mg after approval in Europe) were given, then PRN retreatment for nine months. Individualized re-injection in the trial was based on VA and OCT assessments [44]. The mean change in BCVA compared to baseline was +5.8 and +3.6 ETDRS letters at 3 and 12 months, respectively. This drop between months 3 and 12 (when the PRN regimen was initiated) differs from the 1.3 letter gain in the MARINA and 1.3 letter gain in the ANCHOR studies when a monthly regimen was used. Results showed that the VA in patients reached its maximum after three months of monthly injections. After the PRN regimen injections were initiated, visual acuity dropped slightly over 2–3 months and was maintained throughout the rest of the trial [44].

Other trials evaluated applying a TREX regimen to attempt and reduce the number of office visits. The SALUTE trial followed a Turkish cohort over 12 months to evaluate the efficacy of a TREX regimen. At 12 months, the mean change in the VA was similar between the TREX group and the PRN group. The median change in BCVA (logMAR) was −0.18 and −0.12 in the treat and extend and PRN groups, respectively (*p* = 0.267). While the results indicate no statistically significant difference, treat and extend protocol was associated with a reduction in the number of visits [45]. TREX-AMD is another study that aimed to assess a TREX regimen for treating AMD. In this phase 3 multicenter, randomized controlled trial, patients were randomized to receive either monthly or TREX management. After 12 months mean change in BCVA was +9.2 and +10.5 ETDRS letters (*p* = 0.60) in the monthly and the TREX groups, respectively. The results of TREX-AMD are significant as they provide further evidence that supports a TREX regimen versus monthly dosing when managing patients with nAMD [46].

Along the line of reducing the total number of injections, other trials considered combining anti-VEGF treatments with other treatment modalities. For example, because the effect of PDT is not strictly selective on pathological CNVs, it may cause collateral damage to the adjacent normal choroidal vessels and an increase in the expression of VEGF [47]. Therefore, combining PDT with anti-VEGF injections could serve a dual benefit. Anti-VEGF injections given before or shortly after PDT will neutralize the VEGF produced by the latter. The PDT will decrease CNV perfusion and size, thus decreasing the need for frequent injections and the overall cost of treatment.

The FOCUS study, a phase 1/2, multicenter, randomized, single-masked, controlled study, was conducted on patients with predominantly classic CNV to evaluate the effect of combining PDT with ranibizumab injections. Patients were randomized to receive PTD with monthly sham injections or PDT with monthly 0.5 mg ranibizumab. At 24 months of follow-up, the percentage of patients who had lost fewer than 15 ETDRS letters from baseline VA was 88% in the combined PTD+ranibizumab group versus 75% in the PDT group (*p* < 0.05). Furthermore, patients receiving monthly PDT+ranibizumab required significantly fewer PDT sessions (an average of 0.4) versus 3.0 in the PDT group. While serious intraocular inflammation occurred in 12 % of the PDT+ranibizumab group, it was 0 in the PDT group. While comparing the results of the FOCUS study to the ANCHOR trial, the results of the ANCHOR trial are better. However, the inclusion criteria for the two studies differed. While patients that had received previous PDT treatments were excluded from the ANCHOR study, they were allowed in the FOCUS study. In addition, the ranibizumab formulation used in both studies was different. The FOCUS study used lyophilized ranibizumab, while the ANCHOR trial used a liquid ranibizumab formulation with FDA approval. The increased rate of intraocular inflammation seen in the FOCUS study could be attributed to the formulation used [48]. Subsequently, the PROTECT study [49] combined PDT with 0.5 mg ranibizumab (in its liquid form). Patients included in the study received ranibizumab injections 1 h following PDT. The PROTECT study showed significantly less intraocular inflammation than the FOCUS study, with no patients suffering from vision loss due to intraocular inflammation. Only one patient had mild uveitis (3.1%), while two (6.3%) patients had moderate uveitis.

To further evaluate the efficacy and safety of ranibizumab combined with PDT in nAMD patients, the SUMMIT trial was conducted. The SUMMIT trial consisted of three randomized clinical trials DENALI, MONT BLANC, and EVEREST. 

The DENALI study in the United States and five centers in Canada was a two-year randomized, double-blinded multicenter study [50]. Patients with subfoveal CNV (all angiographic subtypes were included) were randomized to receive either ranibizumab monotherapy, a combination of ranibizumab and standard fluence (SF) PDT, or a combination of ranibizumab and reduced fluence (RF) PDT. The study’s primary objective was to prove the non-inferiority of one of the arms with PDT to ranibizumab monotherapy. Although the primary objective was not met, PDT did reduce the number of ranibizumab injections required. An average of 5.1, 5.7, and 10.5 ranibizumab injections were needed in the SF, RF, and ranibizumab monotherapy groups, respectively [50].

MONT BLANC was a similar study conducted in Europe that compared ranibizumab monotherapy to a combination of SF PDT with ranibizumab. Results at the 12-month follow-up showed non-inferiority of the combination group to the ranibizumab monotherapy group; the safety profile was similar in both groups. It has been reported that specific angiographic subtypes of CNV, such as PCV and retinal angiomatous proliferation (RAP), respond differently to PDT when compared to minimally classic or occult CNVs. Patients with PCV often have a relapsing-remitting course with an overall good visual prognosis, yet half of the patients have persistent leakage and bleeding affecting their visual prognosis [51]. The EVEREST trial was conducted in Asia to compare the efficacy and safety of verteporfin PDT alone, in combination with ranibizumab, vs. ranibizumab monotherapy for symptomatic macular PCV. The trial showed that PDT in combination with ranibizumab or alone was superior to ranibizumab monotherapy regarding complete polyp regression. The proportion of patients with complete regression of polyps at six months was significantly larger in both the PDT combined with ranibizumab group (77.8%) and PDT monotherapy (71.4%) versus ranibizumab monotherapy (28.6%) (*p* < 0.01 for all comparisons) [52]. 

The EVEREST 2 trial was a 2-year trial to evaluate the efficacy of ranibizumab monotherapy vs. ranibizumab with PDT in patients with PCV. The mean visual acuity gain at 24 months was 9.6 letters in the combination therapy group versus 5.5 letters in the monotherapy group (*p* = 0.005). The results highlighted that combination therapy in PCV was superior to ranibizumab monotherapy. The combination group also had increased odds of complete regression of polypoidal lesions (56.6% versus 26.7% of participants; (*p* < 0.001) and fewer overall treatment sessions (median, 6.0 in the combination group) versus (median, 12.0 in the monotherapy group) [53]. Compared to bevacizumab, ranibizumab was found to bind and neutralize VEGF at a lower concentration with “higher efficiency which was found to be maintained for a longer time.” Ranibizumab also had superior potency and retinal penetration [54].

Studies showing adverse events of bevacizumab after intravitreal injection are limited because the drug was not manufactured for this purpose, as opposed to the numerous randomized controlled clinical trials highlighting the safety profile of ranibizumab. Ranibizumab has been used extensively over the past years, and as such, we have real-world results apart from those obtained in a controlled clinical trial setting. However, the real-world results were unlike those set forth by the main clinical trials [55]. Some of these real-world studies are summarized in Table 2. This could be attributed to several reasons; in a real-life setting, we do not have exclusion criteria, and some patients will have co-existent ocular conditions, logistics of obtaining treatment may cause a delay in management, and a significant number of patients may miss follow-up among other factors that may affect both diagnosis and treatment. 

A recent study aimed to assess whether prophylactic ranibizumab can be used in eyes with intermediate AMD changes (multiple intermediate drusen [≥63 μm and <125 μm] or ≥1 large drusen [≥125 μm] and pigmentary changes) and contralateral nAMD. In this multicenter randomized clinical trial, ranibizumab vs. sham injections was given every three months for 24 months. The primary outcome was conversion to nAMD over 24 months, with a change in BCVA from baseline to 24 months as a secondary outcome. Conversion to nAMD over 24 months occurred among 7 of 54 eyes (13%) in both groups (ranibizumab vs. sham, *p* = 0.86). The authors concluded that the prophylactic ranibizumab injections did not seem to prevent conversion to nAMD [71]. 

## 7. Aflibercept

Aflibercept (Eylea; Regeneron, Tarrytown, NY, USA) was approved in 2011. It is a recombinant fusion protein containing specific domains of human VEGF receptors 1 and 2 combined with the Fc portion of human immunoglobulin. Thus, it can target VEGF-A, VEGF-B, and placental growth factor (PIGF) [72].

The (CLEAR-IT) 2 trial was a phase II multicenter, randomized, double-masked clinical trial to assess the effect of intravitreal aflibercept in patients with nAMD. The study consisted of initial 12 weeks with a fixed dosing regimen followed by a PRN regimen in weeks 16 to 52. The primary outcome was a change in CRT, with a change in BCVA as a secondary outcome. There was a significant mean decrease in CRT of 119 μm from baseline in all groups (*p* < 0.0001). At the same time, the BCVA increased significantly by a mean of 5.7 letters at 12 weeks in all groups (*p* < 0.0001) [73]. Two parallel double-masked randomized clinical trials (VIEW-1 and -2) were started in August 2007, comparing intravitreal aflibercept with ranibizumab in the treatment of nAMD [74]. The VIEW-1 study was conducted in North America, while the VIEW-2 was an international study. The study defined maintaining vision as losing less than 15 ETDRS letters. One-year results of the VIEW-1 showed maintenance of vision in 96% of patients receiving 0.5 mg aflibercept monthly, 95% in the group receiving 2 mg monthly, and 95% in those receiving 2 mg every two months. These results were comparable to the 94% achieved in the control group receiving monthly 0.5 mg ranibizumab. Patients receiving aflibercept 2 mg monthly had a better visual gain at 10.9 letters than a mean 8.1 letter gain with monthly ranibizumab 0.5 mg (*p* < 0.01). The VIEW-2 study resulted in similar results with the maintenance of vision achieved in 96% of patients receiving 0.5 mg aflibercept monthly, 96% in the group receiving 2 mg monthly, and 96% in those receiving 2 mg every two months. These results were comparable to the 94% achieved in the control group receiving monthly 0.5 mg ranibizumab. The results of the VIEW-1 and VIEW-2 were comparable to the MARINA and the CATT trials. After the three loading doses of aflibercept, it was dosed every eight weeks, and results remained non-inferior to ranibizumab at a dosing every four weeks in terms of both the efficacy and safety profile. This is an important advantage for aflibercept as it will decrease the number of visits, reduce the financial burden and improve patient compliance. The integrated results from both VIEW-1 and VIEW-2 studies at 52 weeks showed noninferiority of all three aflibercept treatment regimens compared to the ranibizumab regimen [74].

Real-life results aim at establishing a treatment protocol that optimizes the number of treatment visits to decrease the financial burden on the healthcare system and improve patient compliance while maintaining visual outcomes. A study by Rodriguez and colleagues attempted to compare real-life results of aflibercept and ranibizumab in patients with nAMD. This was a retrospective review of patients with nAMD that were treatment naïve and receiving a fixed dosing regimen of either aflibercept or ranibizumab. At the 12-month follow-up, there was not a statistically significant difference in the change in BCVA between the two groups (*p* = 0.121), but the change in CRT was significantly better in the aflibercept group (−142.2 versus −51.5, *p* = 0.011), showing that while visual results were comparable between both groups, the anatomical results were better with aflibercept [75]. Another study by Luska and colleagues retrospectively compared the efficacy of intravitreal aflibercept to ranibizumab in patients with nAMD. Patients in both groups were treated with monthly 0.5 mg ranibizumab and 2 mg aflibercept for three months, followed by four doses at a bimonthly interval and a PRN regimen in the second year. At one year, the change in BCVA was better in the aflibercept group at 9.3 ETDRS letters (*p* < 0.01), while it was 4.3 letters in the ranibizumab group (*p* < 0.01). Further analysis showed that the differences between the aflibercept and ranibizumab groups in terms of BCVA were statistically significant (*p* < 0.01), with aflibercept being superior. The changes in CRT did not differ significantly between the two treatment groups (*p* > 0.05) [76]. Another retrospective review aimed to assess the 4-year results of the anatomical and visual outcomes of intravitreal aflibercept in patients with nAMD while on a treat and extend protocol. BCVA improved significantly from 63.9 ±  15.0 at baseline to 67.3  ±  14.0 at four years (*p*  <  0.01). After four years of treatment, 24% of eyes showed improvement in BCVA ≥15 letters, and 6% of eyes worsened by ≥15 letters compared with baseline. These results display that aflibercept maintains good visual outcomes in nAMD with a treat and extend TREX protocol after four years of follow-up in a real-life setting [77]. While pivotal studies have displayed noninferiority of bimonthly aflibercept to monthly ranibizumab in maintaining vision in patients with nAMD [78], this fixed dosing regimen is less sustainable than the more flexible TREX regimens and PRN regimens.

Another retrospective case series aimed to identify populations with anatomical and functional worsening after switching to bimonthly aflibercept injections and to further investigate whether alternating injections with another anti-VEGF agent would be sufficient for visual outcomes. All patients initially had received a loading dose of 3 monthly injections of aflibercept, patients who showed worsening in retinal fluid status or any hemorrhage after switching into bimonthly injections were switched to receive alternating bevacizumab (1.25 mg, 0.05 mL) with the bimonthly aflibercept. The other patients maintained the bimonthly aflibercept regimen. After 12 months of follow-up, a treat and extend regimen was adopted. Among the initial 72 eyes, 24 (33.3%) showed worsening of retinal fluid after increasing treatment interval to 2 months and were switched to alternating bevacizumab injections. While the mean BVCA improved from baseline and remained steady in both groups, no significant difference was observed between the two groups during follow-up. The CRT decreased from baseline and remained stable through follow-up in both groups. The bimonthly aflibercept group maintained a higher percentage of dry macula than the alternating aflibercept/bevacizumab group throughout the study period. Additional bevacizumab injections could not fully restore the anatomical worsening that occurred in some patients after switching to the bimonthly regimen. This study showed that almost 1/3 of patients with nAMD suffer worsening when switching to bimonthly aflibercept injections after the initial loading phase. It also highlighted that bevacizumab could be used alternately during this period to prevent deterioration of anatomical and visual results [78].

Another retrospective study aimed to evaluate the role of aflibercept in nAMD patients that develop resistance to repeated bevacizumab/ranibizumab injections. Patients with persistent intraretinal or subretinal fluid (IRF/SRF) for at least three months based on OCT findings while on monthly ranibizumab/bevacizumab injections were switched to aflibercept. The mean change in BCVA from baseline in logMAR was 0.05 ±  0.13 (*p*  =  0.01) at month 1, 0.04  ±  0.16 (*p*  =  0.08) at month 3, 0.01  ±  0.22 (*p*  =  0.9) at month 6, and 0.02  ±  0.28 (*p*  =  1) at one year. The mean change in central macular thickness from baseline was 64  ±  75 μm (*p*  <  0.0001) at month 1, 42  ±  85 μm (*p*  =  0.002) at month 3, 47  ±  69 μm (*p*  <  0.0001) at month 6, and 46  ±  99 μm (*p*  =  0.001) at one year. This study showed that aflibercept could be used as a viable option for nAMD patients showing resistance to repeated bevacizumab/ranibizumab injection. It can significantly improve VA in the short term and maintains anatomical improvement during follow-up [79]. 

A randomized single-masked clinical trial aimed to evaluate the efficacy of aflibercept as prophylaxis to prevent conversion to nAMD in high-risk eyes. High-risk eyes were defined by the presence of >10 medium drusen (≥63 to <125 μm), at least one large drusen (≥125 μm), and retinal pigmentary changes, and nAMD in the fellow eye. Patients were randomized to receive prophylactic aflibercept injection versus sham injection every three months. By month 24, 6 patients (9.5%) in the aflibercept group and seven patients (10.9%) in the sham group developed nAMD (*p* =  0.98). This study showed that a prophylactic quarterly aflibercept injection did not have a protective effect in preventing conversion to nAMD. Thus, careful clinical monitoring of fellow eyes in patients with nAMD is required, along with further studies to explore reasons that possibly cause the conversion to nAMD and possible prophylactic options [80].

## 8. Brolucizumab

Brolucizumab is a novel, newly FDA-approved (October 2019) anti-VEGF agent used to treat nAMD. It’s a humanized single-chain fragment antibody composed of 255 amino acids. It binds all forms of VEGF-A, reducing permeability and neovessel formation. Its high solubility and concentration properties facilitate delivery with a binding capacity 11 or 22 times higher than other anti-VEGF agents. 

The HAWK and HARRIER studies were two similarly designed phase III, double-masked, multicenter, randomized controlled trials comparing brolucizumab to aflibercept in the treatment of nAMD. Patients were randomized 1:1:1 to receive either brolucizumab 3 mg, brolucizumab 6 mg, or aflibercept 2 mg (HAWK study) or 1:1 of either brolucizumab 6 mg or aflibercept 2 mg (HARRIER study). The loading phase consisted of injections at weeks 0, 4 and 8. Brolucizumab was injected after every 12 weeks (q12w) or every eight weeks (q8w) if disease activity warranted less spacing out of injections in a regimen known as q12w/q8w. Aflibercept was injected q8w [81]. In the HAWK trial, patients demonstrated noninferiority of aflibercept in mean change in BCVA from baseline (+6.1 letters in brolucizumab 3 mg, +6.6 letters in brolucizumab 6 mg, and +6.8 letters in the aflibercept 2 mg groups) (*p* < 0.001). The HARRIER study also demonstrated noninferiority of aflibercept in mean change in BCVA from baseline (+6.9 letters in brolucizumab 6 mg group and +7.6 letters in the aflibercept 2 mg groups) (*p* < 0.001). 56% (HAWK) and 51% (HARRIER) of brolucizumab 6 mg treated eyes were maintained on the q12w regimen without the need to revert to q8w through weeks 48. Intraretinal and subretinal fluids at 16 and 48 weeks were less frequent in brolucizumab-treated eyes compared to aflibercept. With the following results for the HAWK study at 48 weeks (3 mg brolucizumab, 34.1% versus 44.7% in aflibercept group; *p* = 0.002) and (6 mg brolucizumab, 31.2% vs. 44.6% in aflibercept group; *p* < 0.001) and (25.8% vs. 43.9%; *p* < 0.001) in the HARRIER study.

The SWITCH study revealed real-world short-term outcomes for patients with nAMD that were poorly responsive to other anti-VEGF agents and then were switched to brolucizumab. BCVA was measured four weeks after the first dose of brolucizumab. The mean change in BCVA was 0.03 ± 0.14 logMAR (*p* = 0.115). There was also a reduction in CRT with a mean reduction of −66.76 ± 60.71 µm for CRT and (*p* < 0.001) [82]. The PROBE study evaluated the effect of the PRN brolucizumab dosing regimen without an initial loading dose in patients with nAMD. A retrospective, multi-center, observational study included 27 treatment-naïve patients nAMD patients that received a PRN brolucizumab regimen with spacing of at least eight weeks. BCVA changed from a mean of (57.4 ± 4.5 ETDRS letters) during the initial visit to (65.3 ± 3.1 letters; *p* = 0.014) during the final follow-up visit. There was a mean gain of 7.8 ± 3.5 ETDRS letters. Change in CRT decreased significantly from 398.1 ± 47.2 μm to 283.0 ± 57.2 μm at the final follow-up visit (*p* = 0.021) [83]. The REBA study was a retrospective, multi-center, observational study that included 78 nAMD patients (some were treatment-naive, and others were “switch” patients). “Switch” patients were switched from other anti-VEGF therapies due to either recurrence, resistance, or inability to treat and extend without deterioration in outcomes. The mean change in BCVA was +11.9 ± 3.9 letters (*p* = 0.011) and +10.4 ± 4.8 letters (*p* = 0.014) in the treatment-naïve and switch groups, respectively [84].

While brolucizumab was generally well tolerated compared to aflibercept in the HAWK trial, there was a higher rate of intraocular inflammation. Uveitis and iritis developed in 2.2% of brolucizumab-treated patients compared to 0.3% (uveitis) and 0 (iritis) in the aflibercept group. Most cases of uveitis and iritis (90%) were mild to moderate and treated with topical medications for a short period with no sequelae [82]. However, recent reports on the risk of retinal vasculitis with intravitreal brolucizumab injection should be further evaluated. Particular caution must be taken when using brolucizumab in patients with a history of intraocular inflammation, retinal vasculitis, eyes with scleritis and episcleritis, and with a history of culture-negative endophthalmitis [85]. The SWITCH study showed seven eyes of seven patients with intraocular inflammation, including one with retinal vasculitis [82]. In the REBA study, one patient had a macular hole, and one developed vascular occlusion (both recovered without sequelae) [84].

## 9. Newer Anti-VEGF Medications

Ranibizumab has proven its efficacy in the first pivotal clinical trials, but subsequent real-life studies showed a discrepancy in the results obtained compared to previous controlled clinical trials. Currently, the rationale behind newer treatment modalities is to provide a more durable treatment to address the problem of noncompliance related to repeat treatments and frequent follow-up visits.

## 10. Ranibizumab Portal Delivery System

The ranibizumab portal delivery system (RPDS) (Susvimo™) consists of an implantable re-fillable reservoir that requires surgical implantation. This reservoir can be re-filled on an as-needed basis in the office, offering a practical solution to sustain the visual outcomes [86]. The phase 1 trial of RPDS was an open-label prospective study conducted on 20 nAMD treatment naïve patients. Primary outcomes were to assess safety (incidence and frequency of adverse events) and problems related to implantation, refill, and explanation. Secondary outcomes were improvement in BCVA and anatomical outcomes. Conjunctival hyperemia occurred in 95% of patients, vitreous hemorrhage in 25% of patients, and hyphema in 20%. There were four reported serious adverse events (endophthalmitis, two cases of persistent vitreous hemorrhage, and traumatic cataract). Approximate improvement in BCVA was 10 ETDRS letters in the 20 patients at 12 months [87]. An animal model was used to assess the surgical technique to determine the cause of vitreous hemorrhage. The only surgical parameter that reduced the incidence of this adverse effect was pars plana hemostasis before pars plana incision. Thus, a technique modification entailed edge-to-edge laser photocoagulation at the pars plana to achieve adequate hemostasis.

The phase 2 trial (LADDER) was a randomized, multi-center, interventional, controlled trial [86]. Patients were randomized to receive 10, 40, 100 mg/mL RPDS or 0.5 monthly ranibizumab injections. The primary outcome in patients implanted was refill time. In the 10, 40, and 100 mg/mL RPDS groups, the median time to first implant refill was 8.7 months, 13.0 months, and 15.8 months, respectively. The patients who did not require a refill for the first six months or more were 63.5%, 71.3%, and 79.8%, respectively. Regarding visual outcomes, the adjusted mean BCVA from baseline in the 10, 40, and 100 mg/mL and 0.5 mg of monthly ranibizumab groups were −4.6 ETDRS letters and −2.3 ETDRS letters, and +2.9 ETDRS and +2.7 ETDRS letters, respectively at 22 months [88]. The phase III (ARCHWAY) study was a randomized, multicenter, open-label comparative study comparing the efficacy, safety, and pharmacokinetics of the RPDS with monthly 0.5 mg ranibizumab in patients with nAMD. 94.8% of patients in the RPDS went six months without additional interventions and were non-inferior to the ranibizumab group. 57.8% of patients gained vision in the RPDS group vs. 58.9% in the ranibizumab group. The visual gain in ETDRS letters from baseline, when averaged over 36–40 months, was 0.2 letters in the RPDS group and 0.5 letters in the ranibizumab group [89]. The PORTAL study is a multi-center, non-randomized, open-label extension study for nAMD patients who have completed the LADDER or the ARCHWAY studies. The study aimed to evaluate the long-term safety and tolerability of 100 mg/mL RPDS with refills administered every 24 weeks for 144 weeks. The results of this study which have not yet been published will evaluate the incidence and the severity of ocular and systemic adverse events, incidence duration and severity of adverse events of special interests (AESI), and AESI-related to the RPDS in the postoperative period. Secondary outcomes will assess BVCA in the study population [90].

A cost analysis was conducted to evaluate and compare the costs of RPDS versus intravitreal Anti-VEGF injections in patients with nAMD. The authors demonstrated that the mean number of intravitreal injections to break even with the cost of one PDS with one refill was 10.8, 9.3, and 34.5 injections of ranibizumab, aflibercept, and bevacizumab, respectively. RPDS with fixed 6-months refills over a one-year duration cost $21,016. The monthly intravitreal injections of ranibizumab cost $1943 more (*p* = 0.34), aflibercept cost $5702 more (*p* = 0.04), and bevacizumab cost $16,732 less (*p* < 0.001) [91]. 

A study was conducted to evaluate patient satisfaction with the RPDS and how it compared to monthly injections for their management. Patients with nAMD were randomized to receive either RPDS with fixed refill exchanges every 24 weeks or ranibizumab injections every four weeks. Treatment satisfaction was measured using the Macular Disease Treatment Satisfaction Questionnaire in the RPDS and the ranibizumab injection groups at 40 weeks. Patient preference and satisfaction were measured using a validated PDS Patient Preference Questionnaire (PPPQ) which aimed to measure the proportion of patients using the RPDS that preferred it over previous intravitreal injections or current injections in the other eye at week 40. Treatment satisfaction scores in both groups were high, with differences between the two groups being minimal but in favor of the RPDS treatment (difference, 1.9; 95% CI, 0.7–3.1). Patient preference and satisfaction scores showed that most patients assigned to RPDS treatment preferred it over previous intravitreal injections (93.2%). Of these patients (73.5%) strongly preferred the PDS. These findings support the hypothesis that PDS would prove convenient to patients and eventually improve compliance and results [92].

## 11. Abicipar Pegol

Abicipar (AGN-150998, MP0112, abicipar; Allergan plc/Molecular Partners) is a specific protein binding molecule that binds with high affinity to soluble isoforms of VEGF-A [93]. Compared to ranibizumab, abicipar has a smaller molecular weight (34 kDa vs. 48 kDa), higher binding affinity, and a longer intraocular half-life time (>13 days in the aqueous humor versus seven days in the aqueous humor) [94]. These properties are thought to give abicipar more durability than the currently available anti-VEGFs. Abicipar was evaluated in a phase II study REACH. This study was conducted in three stages. The first stage aimed to assess the abicipar following a single intravitreal injection of abicipar in patients with nAMD. Stage 2 compared and evaluated the safety and efficacy of ranibizumab and abicipar when used at a PRN dosing in treatment-naïve nAMD patients. REACH 3 was intended to compare safety, efficacy, and the pharmacokinetic profiles of abicipar 1-mg, 2-mg when compared to 0.5 mg ranibizumab. Furthermore, to evaluate the durability of abicipar if used at eight or 12-week dosing intervals. Study visits were scheduled at baseline, day 3, and weeks 1, 4, 8, 12, 16, and 20. Patients were randomized to receive either abicipar 1-mg, 2-mg, or 0.5 mg ranibizumab. Efficacy evaluations were based on BCVA and CRT using OCT measurements. There were improvements in BCVA in all arms of the study, but no statistically significant difference was present between the abicipar and the ranibizumab arms. At week 20 (12 weeks after the last injection of abicipar and four weeks after the last injection of ranibizumab), the mean change in BCVA from baseline was +8.2, +10.0, and +5.3 ETDRS letters in the abicipar 1 mg, abicipar 2 mg, and ranibizumab 0.5 mg arms, respectively. At week 20, the proportion of patients achieving ≥15-ETDRS letters were 14.3%, 15.4%, and 14.3%, in the abicipar 1 mg, 2 mg, and ranibizumab groups, respectively (no statistically significant difference between groups). The mean CRT reduction from baseline was 116, 103, and 138 μm at week 20 in the abicipar 1 mg, abicipar 2 mg, and ranibizumab 0.5 mg arms, respectively (no statistically significant difference between the groups). Overall, abicipar was well tolerated the number of treatments related to adverse events was comparable to ranibizumab. Thus, the REACH 3 study demonstrated similar results regarding BCVA and CRT improvements with five ranibizumab injections vs. 3 abicipar injections [93].

Two phase III multi-center, randomized clinical trials with identical designs (CEDAR and SEQUOIA). These studies were intended to study the safety and efficacy of abicipar q8w and q12w compared to ranibizumab in patients with nAMD. At enrollment, patients were randomized to receive either abicipar 2 mg q12w after initial baseline and week four injections, abicipar 2 mg q8w after initial baseline and week four injections, or ranibizumab 0.5 mg every four weeks. At week 104, after 14, 10 and 25 injections in the abicipar q8, abicipar q12, and ranibizumab groups, the mean change in BCVA from baseline was +7.8 letters, +6.1 letters, and +8.5 letters, respectively [95]. The proportion of patients with ≥15 letter gain at 104 weeks was assessed. The difference in the proportion of patients between the abicipar and ranibizumab groups was 1.1% (95.1% CI, −4.9% to 7.1%) for the abicipar q8 group and −4.6% (95.1% CI, −10.4% to 1.2%) for the abicipar q12 group. The improvement in CRT measurements was maintained through the 104 weeks in all treatment groups. The overall results revealed noninferiority of abicipar q12 compared to ranibizumab injections in managing patients with nAMD [96]. The overall incidence of adverse events, including ocular ones, was similar in all treatment groups. Yet, the incidence of drug-related adverse events was higher in the abicipar group, namely intraocular inflammation. In CEDAR, intraocular inflammation occurred in 15.1% of patients on the q8w arm and 15.4% in the q12w groups, compared with 0 in the ranibizumab group. In SEQUOIA, inflammation occurred in 15.7% and 15.3%, and 0.6 % of the q8w, q12w, and ranibizumab groups, respectively. To reduce the adverse effects of abicipar, a new modified formulation with reduced impurities was introduced in a new phase 2 trial, the MAPLE study. This open-label single-arm trial used the new formulation. Patients with treatment naïve nAMD received three monthly loading doses. Intraocular inflammation occurred in 8.9%, with 1.8% being severe [97].

## 12. Faricimab

Faricimab (Vabysmo), developed by Roche/Genentech, is a bispecific antibody that functions through independent and simultaneous binding and inhibition of both angiopoietin-2 (Ang-2) and VEGF-A. Ang-2 has a vital role in inflammation and vascular destabilization. Thus, neutralizing Ang-2 may restore vascular stability and decrease leakage, inflammation, and neovascularization. It also enhances the response to VEGF inhibition; therefore, dual pathway inhibition is thought to provide more durable results for nAMD patients [98,99].

TENAYA and LUCERNE were identically designed phase 3 multicenter, randomized, double-masked clinical trials. They aimed to assess and evaluate the dual-pathway inhibition provided by faricimab in patients with nAMD. Eligible patients enrolled in the study were randomized in a 1:1 ratio to receive either faricimab up to every 16 weeks or aflibercept every eight weeks [100]. TENAYA and LUCERNE met their primary endpoints with noninferiority to aflibercept in mean change of BCVA at the primary endpoint visits with faricimab dosed up to 16 weeks and aflibercept every eight weeks. In the TENAYA study, adjusted mean change in BCVA was 5.8 letters in the faricimab group and 5.1 letters in the aflibercept group (95% CI −1.1 to 2.5). In the LUCERNE study, the mean change in BCVA was 6.6 letters in the faricimab group and 6.6 letters in the aflibercept group (95% CI −1.7 to 1.8). Adjusted mean change in central subretinal thickness (CST) from baseline to the primary endpoint visits was −136.8 μm with faricimab and −129.4 μm with aflibercept in TENAYA, and −137.1 μm with faricimab and −130.8 μm with aflibercept in LUCERNE. 

In the TENAYA and LUCERNE studies, the overall incidence of ocular adverse events was similar in both treatment groups. TENAYA and LUCERNE are ongoing clinical trials, with the results of the second year and the long-term follow-up results yet to be published. The studies proved the non-inferiority of faricimab to aflibercept but considering the treatment regimen that can be extended up to 16 weeks versus the eight weeks for aflibercept makes faricimab a more efficient treatment option [100].

## 13. KSI-301

KSI-301, developed by KODIAK sciences, Palo Alto, CA, is an antibody conjugate formed of anti-VEGF monoclonal antibody phosphorylcholine-based polymer to enhance stability and increase half-life in the eye. This design aimed to provide a higher concentration in the eye without the same bioactivity compared to the current standard of care.

NCT03790852 is a phase 1, randomized, open-label study to compare the safety, tolerability, and pharmacokinetics of 2 doses of KSI-301 in 35 nAMD patients, 35 diabetic macular edema patients, and 35 retinal vein occlusion patients (n = 35). Patients are randomized to receive either a 2.5 or 5 mg dose of KSI-301 [101]. The mean change in BCVA at week 20 was +4.3 letters in nAMD, and the mean change in CST was −67µm in nAMD. At 24 weeks, 55% of patients were able to extend six months before required re-treatment. 

There were no reported intraocular side effects or other significant side effects. Preliminary results of the trial have shown promising functional and anatomical outcomes [101]. Similar to the NCT03790852 study, The DAZZLE study is a multi-center, double-masked clinical trial; after three initial loading doses, patients in the aflibercept group had monthly injections while those in the KSI-301 group had injections every 3–5 months. This study’s primary efficacy, safety, and durability results have not been published yet [102].

## 14. OPT-302

OPT-302 was developed by OPHTHEA limited. It inhibits the activity of VEGF C [103]. Results of animal studies have shown the superior combined effect of OPT-302 and ranibizumab versus ranibizumab alone. In a phase IIb randomized, multi-center, double-masked interventional clinical trial, patients with nAMD were randomized into three groups to receive either 2 mg of OPT-302 with ranibizumab 0.5 mg (Group A), 0.5 mg of OPT-302, and 0.5 mg of ranibizumab (Group B), or 0.5 mg or ranibizumab alone (Group C). At 24 weeks, the mean change in BCVA was 14.22 letters for Group A, 9.44 letters for Group B, and 10.84 letters for Group C. Group A demonstrated superiority in BVCA gain of +3.4 letters (*p* = 0.0107) compared with Group C. The change in the total lesion area from baseline was −4.33 mm^2^ for Group A compared with −3.11 mm^2^ for Group C (*p* = 0.0137). The overall safety profile was comparable between the three arms of the study [104]. Two parallel phase 3 studies are currently in process; the study of OPT-302 in combination with Ranibizumab (ShORe) and Combination OPT-302 with Aflibercept Study (COAST) has begun [104].

## 15. GB-102

Like RPDS, the GB-102 is another anti-VEGF sustained release delivery system; GB-102 is designed as an intravitreal formulation of sunitinib malate-containing, biodegradable microparticles. Sunitinib malate is a tyrosine kinase inhibitor that targets both VEGF-A and PDGF. After injection, it forms a depot in the vitreous, allowing controlled and sustained release; thus, it’s intended to maintain visual and anatomical outcomes for six months before another dose is required.

ADAGIO NCT03249740 is a phase 1/2 open-label, multi-center study on GB-102 in patients with nAMD. 4 cohorts of patients received escalating GB-102 doses either 0.25, 0.5, 1 or 2 mg. Primary endpoints regarding the overall safety and tolerability of the drug were met. Secondary endpoints addressed the stability of VA and retinal thickness. 88% of the patients at three months and 68% at six months were maintained using a single dose of GB-102. OCT measurements showed a reduction of CST at all months when compared with pre-treatment results (*p* < 0.05). The two mg-arm had dispersion of microparticles into the anterior chamber with deterioration in VA. The phase 2b ALTISSMO NCT03953079 study evaluated the efficacy of GB-102 in patients with nAMD. Patients were randomized to receive either 1 mg GB-102 every six months, 2 mg GB-102 every six months, or 2 mg aflibercept every two months. The primary outcome is the number of patients who will remain rescue free after ten months [104].

## 16. RGX-314

RGX-314 was manufactured by Regen BioPharma (La Mesa, CA). It uses a novel vector to deliver a genome that subsequently induces the production of an anti-VEGF Fab, like ranibizumab. It is delivered during vitrectomy via subretinal injection. This therapy is intended to be a one-time gene therapy for patients with nAMD. ATMOSPHERE, an open-label, multi-center trial aimed to evaluate the safety and efficacy of escalating doses of RGX-314. All patients were initially given a ranibizumab dose to ensure adequate response. Patients were divided into five arms (cohorts) of the study and given their respective doses of RGX-314; rescue injections were given to patients with increased disease activity. At 26 weeks, the primary endpoint of safety and tolerability of RGX-314 has been met. At 1.5 years, patients in the two higher dose cohorts (4 and 5) demonstrated stable VA with a mean change in BCVA change of +1 letters and −1 letters from baseline and decreased CRT, with a mean change of −46 and −93 µm, respectively. At 1.5 years, 4 out of 12 (33%) patients in cohort 4 did not require anti-VEGF treatment six months after RGX-314 administration, with a mean change in BCVA from baseline of +2 letters.

The promising results of the ATMOSPHERE study led to the AAVIATE study. AAVIATE is an open-label trial that will evaluate the safety, efficacy, and tolerability of suprachoroidal delivery of RGX-314 using the SCS Microinjector^®^. The first two cohorts of patients were randomized to receive either two escalating doses of “2.5 × 10^11^ and 5 × 10^11^ genomic copies per eye (GC/eye)” of RGX-314 versus monthly ranibizumab 0.5 mg at a ratio of 3:1. The third cohort received the same RGX-314 dose as the second cohort in patients who are positive for neutralizing antibodies. Results from November 2021 showed that RXGX-314 was generally well tolerated in all three cohorts. Cohorts 1 and 2 showed stable BCVA and CRT at six months. (29%) and s (40%) in Cohorts 1 and 2, respectively, did not require Anti-VEGF treatments six months after RGX-314 administration, resulting in a meaningful reduction in anti-VEGF treatments (>70%) [105]. RGX-314 presents a promising role for gene therapy in the treatment of nAMD patients [105].

## 17. PAN-90806

PAN-90806, from PanOptica (Mount Arlington, NJ), is. A tyrosine kinase inhibitor of VEGF-A and PDGF. It is a topical drop, hypothesized to travel via a transscleral vascular route to reach its target tissue in the retina. In a phase 1/2, multi-center, double-masked clinical trial, patients received escalating doses of PAN-90806 drops. Patients received the drops daily for 12 weeks and were evaluated weekly for safety and the need for rescue treatments. 51% of patients completed the study without the need for rescue treatments. 88% of the non-rescued patients showed clinical improvement or stability. 17.6% of patients had at least one drug-related adverse event, of which almost half were cornea related. Patients had punctate keratopathy due to off-target inhibition of the corneal epithelial epidermal growth factor receptor [106]. Nonetheless, these drops offer a promising non-invasive topical treatment as a possible monotherapy for patients with nAMD.

## 18. ICON-1

ICON-1, from Iconic Therapeutics (South San Francisco, CA, USA), is a recombinant modified factor VIIIa linked to the Fc portion of a human immunoglobulin G1. It binds to tissue factor, which is found to be overexpressed in nAMD while not affecting blood coagulation; the Fc portion of ICON-1 is designed to bind to the Fc receptor of natural killer cells inducing antibody-dependent cellular toxicity to reduce nAMD. 

A phase 1 multi-center dose escalation trial NCT03452527 evaluated intravitreal injection of ICON-1 in patients with CNV due to nAMD. Primary endpoints of safety and tolerability were met with no reported serious side effects [107]. EMERGE-2 is a phase 2 trial in which patients were randomized to receive either 0.5 mg ranibizumab and 0.3 mg of ICON-1, 0.5 mg of ranibizumab only, or 0.3 mg of ICON-1 only. All patients received monthly injections, after which they received injections PRN. Patients in the ICON-1 monotherapy group had only a mild reduction in CRT and stable BCVA. At six months, the combination patients had a 40% reduction in CNV versus 17.2%in the ICON-1 versus 14.6% in the ranibizumab monotherapy arms. Improvement in BCVA (+8.4 letters in the vs. +8.3 letters), as well as a decrease in CRT (−83.9 vs. −91.4 μm) in the combination and ranibizumab groups, were comparable [108].

DECO is a phase 2 randomized, multi-center open-label study in patients with CNV secondary to nAMD. All patients will receive an initial aflibercept injection followed by maintenance with either 0.6 mg ICON-1 or 2 mg aflibercept. The primary endpoint of the study is a change in the size of the CNV after nine months secondary endpoint will be changes in BCVA and the extent of treatment-free intervals [109].

## 19. Conbercept

Conbercept (KH902), by Chengdu Kanghong Pharmaceuticals Group Co., Ltd. (Chengdu, China). Conbercept binds to VEGF-B, many isoforms of VEGF-A, and PGF. It is a fusion protein formed of domains from VEGFR-1 and VEGFR-2 fused to the Fc portion of human IgG. PHOENIX, a phase 3, multicenter, double-masked, controlled clinical trial, was conducted to evaluate the efficacy of Conbercept in managing patients with nAMD. Patients were randomized to Conbercept or sham groups. For 12 months, the Conbercept group received three monthly injections of Conbercept, followed by an injection every three months. In contrast, the sham group received three monthly sham injections followed by three monthly conbercept injections followed by injections every three months. The primary endpoint was the mean change in BCVA at month “3”. In the third month, the mean change in BCVA was +9.20 letters versus +2.20 ETDRS letters in the conbercept and sham groups, respectively (*p* < 0.0001). At the 12-month mark, there was no statistically significant difference in BCVA improvements between the two treatment groups.

This trial gave Conbercept approval to be used in China. Conbercept was undergoing phase 3 clinical trials in the US, PANDA1, and PANDA2, which were terminated in April 2021 as they had not met the primary endpoint. The global pandemic limited the ability of many recruited patients to complete their follow-ups.

## 20. Conclusions

Neovascular AMD continues to be one of the leading causes of visual disability in the elderly population. While treatment outlines have shifted and evolved over the years with the introduction of Anti-VEGF agents, it comes with the cost of frequent injections and visits. The financial burden imposed on the healthcare system and patients hinders the sustainability of treatment and adversely affects treatment outcomes. New agents are being developed to reduce the required treatments while maintaining or improving previous results. While many of these agents have shown promising preliminary results, they are yet to face the test of time and prove noninferiority to the current treatment modalities.

## Figures and Tables

**Figure 1 biomolecules-12-01629-f001:**
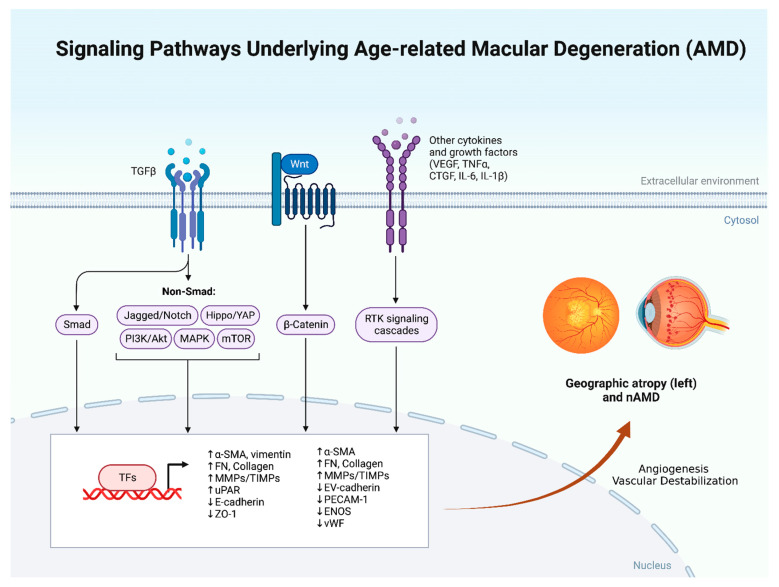
Summary of the pathophysiologic mechanisms involved in the development of age-related macular degeneration.

**Figure 2 biomolecules-12-01629-f002:**
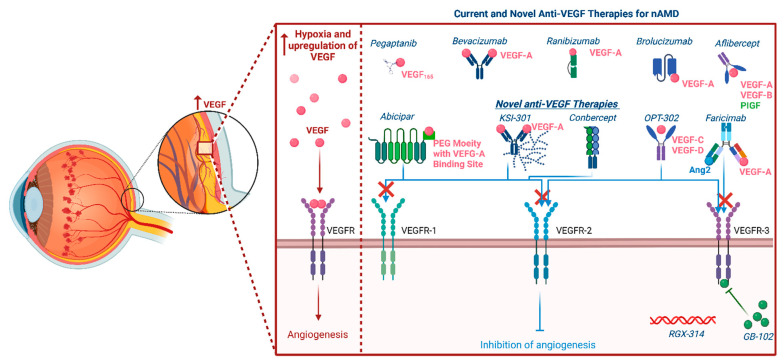
Illustrative diagram summarizing different pharmaceutical modalities for the management of neovascular age-related macular degeneration and their mechanisms of action.

**Table 1 biomolecules-12-01629-t001:** Summarizes the important studies evaluating the safety and efficacy of photodynamic therapy (PDT) in neovascular age-related macular degeneration (nAMD).

Study	Main Study Objective	Study Design	Interventions	Results
Treatment of AMD with PDT (TAP studies) [20]	Evaluation of efficacy of PDT in nAMD patients	Two multicenter, double-masked, randomized, controlled studies, in Europe and the United States of America	Patients were randomized to receive either PDT or placebo	The primary endpoint was the percentage of eyes that lost less than 15 ETDRS letters from baseline at 12 and 24 months. PDT was significantly better than placebo at 12 months (61% versus 46%) And 24 months (53% versus 38%) (*p* < 0.001)
Verteporfin in PDT (VIP) studies [21]	Evaluation of the safety and efficacy of PDT in patients with occult lesions.	Multicenter, randomized, double-masked, controlled clinical trial	Patients were randomized to receive either verteporfin or a placebo	Results at 12 months were disappointing but efficacy was noted at 24 months. At 24 months verteporfin-treated eyes were less likely to have a moderate or severe visual loss, 30% versus 47 % verteporfin-treateded eyes and placebo-treated eyes, respectively, lost at least 30 letters (*p* = 0.001)
Verteporfin in Minimally Classic Choroidal Neovascularization (CNV) (VIM studies) [22]	Evaluation of the efficacy of PDT in minimally classic membranes below six-disc areas in size	Phase 2 Multicenter, randomized, double-masked, controlled clinical trial	Patients were randomized to receive verteporfin with reduced fluence (RF) or standard fluence (SF) or a placebo infusion with either RF or SF	At 24 months, the loss of at least three-lines occurred in 26% of the RF group (*p* = 0.003), 53% of the SF group (*p* = 0.54), and 62% of the placebo group (*p* = 0.03) At 24 months, progression to minimally classic CNV was more common in the placebo group 28%, versus the RF group (5%) (*p* = 0.007) and versus the SF group (3% with *p* = 0.002)
Visudyne in Occult Classic Choroidal Neovascularization (VIO study) [23]	Evaluation of the role of PDT in patients with an occult lesion (lacking any classic component)	Multicenter, randomized, double-masked, controlled clinical trial	Patients were randomized to receive either verteporfin PDT or placebo	There were no statistically significant differences in the degree of vision loss in both the PDT and the placebo groups at 12 and 24 months.

TAP: Treatment of Age-Related Macular Degeneration with Photodynamic Therapy; ETDRS: Early Treatment Diabetic Retinopathy Study.

**Table 2 biomolecules-12-01629-t002:** Summary of real-world studies evaluating the efficacy of ranibizumab in the management of neovascular age-related macular degeneration (nAMD).

Study	Number of Patients/Country	Duration	Objective	Study Design	Treatment	Prior Treatments	Visual Acuity
Chavan et al. [56]	123 eyes in 120 patients United Kingdom	3 years	To describe bilateral visual outcomes after treatment, and the effects of incomplete follow up	Retrospective data collected over 36 months from consecutive patients over 9 months	3 monthly injections of ranibizumab 0.5 mg followed by a pro re nata (PRN) dosing regimen	Naïve patients	Mean change in visual acuity (VA) from Baseline ETDRS letters −1.68 ± 17.76
Cohen et al. [57] (LUMIERE)	551 eyes in 551 patients France	1 year	To survey compliance with recommended intravitreal ranibizumab treatment protocols in daily clinical practice in France, with reference to outcomes.	Retrospective, descriptive observational study. Data on patients were collected after 12 months of treatment with ranibizumab	3 monthly injections of ranibizumab 0.5 mg followed by a PRN dosing regimen	Naïve patients	Mean change in VA from Baseline ETDRS letters +3.2 ± 14.8
Finger et al. (WAVE) [58]	3470 patients Germany	1 year	Evaluation of efficacy and safety of repeated ranibizumab 0.5 mg injections	Prospective non interventional study including AMD patients from 274 practices over a defined period	3 monthly injections of ranibizumab 0.5 mg followed by a PRN dosing regimen	Both naïve and previously treated were included	Mean change in BCVA +0.02 ± 0.01 SEM (LogMAR Snellen Values)
Frennesson and Nilson [59]	312 eyes in 268 patients Sweden	3 years	Evaluation of the effect of carrying forward the last VA of dropouts to the first evaluation point (to get more accurate results)	Retrospective data on patients treated with ranibizumab and followed up for 36 months	3 monthly injections of ranibizumab 0.5 mg followed by a PRN dosing regimen	Both naïve and previously treated were included	Change in BCVA ETDRS: at 36 months +0.1 letter. However, if the last available acuity of dropouts was carried forward, VA decreased by 4.1 letters (*p* = 0.003) at 36 months
Gabai et al. [60]	100 eyes in 92 patients Italy	1 year	Evaluation of efficacy and safety profile of ranibizumab	Retrospective data was collected on patients with Ranibizumab treatment and follow-up for neovascular AMD for > 12 months	3 monthly injections of ranibizumab 0.5 mg followed by a PRN dosing regimen	Naïve patients	Mean change in VA from Baseline ETDRS letters −2.0 ± 17.6
Hjelmqvist et al. (Swedish Lucentis Quality Registry) [61]	471 patients (272 retrospectively and 199 prospectively) Sweden	1 year	Evaluation of efficacy of Ranibizumab	12-month, open-label, observational, prospective, and retrospective study of ranibizumab administration for wet AMD	3 monthly injections of ranibizumab 0.5 mg followed by a PRN dosing regimen	Not Mentioned	Mean change in VA from Baseline ETDRS letters +1.0 ± 13.6
Holz et al. (AURA) [62]	2227 patients, multicenter in Canada and Europe	2 years	Evaluation of efficacy ranibizumab in management of wet AMD in a real-life setting	Retrospective non-interventional observational study	Treatment as prescribed by a physician (not all patients received 3 months loading dose)	Not mentioned.	Mean change in VA from Baseline ETDRS letters at 2 years +0.6 letters
Kumar et al. [63]	81 eyes in 81 patients UK	1 year	Evaluation of efficacy ranibizumab in management of wet AMD in a real-life setting	Prospective study following patients starting ranibizumab for wet AMD	3 monthly injections of ranibizumab 0.5 mg followed by a PRN dosing regimen	Naïve patients	Mean change in VA from Baseline ETDRS letters +3.7 ± 10.8
Matsumiya et al. [64]	54 patients, 24 with wet AMD and 30 with PCV Japan	1 year	Evaluation of the efficacy of ranibizumab in management of two types of AMD	Retrospective cohort study	3 monthly injections of ranibizumab 0.5 mg followed by a PRN dosing regimen	Both naïve and previously treated were included	Change in BCVA at 12 months (log MAR values) PCV: −0.04 Typical AMD: −0.16
Muether et al. [65]	102 patients Germany	1 year	Evaluation of the effect of latency and delay in initiation of treatment	Prospective study following patients with wet AMD	3 monthly injections of ranibizumab 0.5 mg followed by a PRN dosing regimen. The German Health System caused a delay of 23.5 ± 10.4 days Between the decision to treat and initiation of treatment	Naïve Patients	Mean change in VA from Baseline ETDRS letters +0.66 ± 16.82
Nomura et al. [66]	123 patients Japan	1 year	Evaluation of the effect of Vitreomacular adhesions VMA during treatment with ranibizumab in patients with wet AMD	Retrospective study	3 monthly injections of ranibizumab 0.5 mg followed by a PRN dosing regimen	Naïve patients	Mean change in VA from Baseline ETDRS letters Absent VMA: +6 letters VMA: +1.5 letters
Pagliarini et al. (EPICOHORT) [67]	755 patients, 133 of which had bilateral treatment Europe	2 years	Evaluation of the efficacy and safety profile of Ranibizumab in a real-life setting	Prospective, Phase 4 observational trial	3 monthly injections of ranibizumab 0.5 mg followed by a PRN dosing regimen	Both naïve and previously treated were included	Mean change in VA from Baseline ETDRS letters At 12 months: +1.5 ± 0.61 (SEM)
Piermarocchi et al [68]	94 eyes of 94 patients Italy	1 year	Evaluation of the effect of genetic and non-genetic factors in treatment response to ranibizumab in wet AMD	Prospective study	3 monthly injections of ranibizumab 0.5 mg followed by a PRN dosing regimen	Naïve Patients	Mean change in VA from Baseline ETDRS letters At 12 months 0.97 ± 9.1
Rakic et al. (HELIOS) [69]	309 eyes in 267 paients Belgium	2 years		Prospective multicenter observational study	3 monthly injections of ranibizumab 0.5 mg followed by a PRN dosing regimen	Both naïve and previously treated were included	Mean change in VA from Baseline ETDRS letters At 24 months −2.4 ± 17.4
Zhu et al. [70]	886 patients 208 eyes of 208 patients completed the study Australia	5 years	Evaluation of the efficacy and safety profile of Ranibizumab in a real-life setting	Retrospective study	3 monthly injections of ranibizumab 0.5 mg followed by a PRN dosing regimen	Both naïve and previously treated were included	Mean change in VA from Baseline ETDRS letters At 5 years: −2.4

ETDRS: Early Treatment Diabetic Retinopathy Study, BCVA: best corrected visual acuity, log MAR: logarithm of the minimum angle of resolution, PCV: polypoidal choroidal vasculopathy.

## Data Availability

Not applicable.

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
