# Peer review of "Current and Novel Therapeutic Approaches for Treatment of Neovascular Age-Related Macular Degeneration"

_biomolecules, 2022, doi:10.3390/biom12111629_

Round 1

Reviewer 1 Report

This is a well-organized review compiled with the latest and ongoing progress of pre-clinical or clinical therapeutic development for treating neovascular AMD, the leading cause of vision damage among AMD patients. I only have a few comments: 

1. reference #3 is not clearly support the statement "Neovascular AMD (nAMD) is responsible for 90% of severe vision loss and blindness caused by AMD" (line 43). Please add more supporting references.

2. The title of Figure 1 is incomplete

3.  Line 863, ßIn a phase ½ randomized, change to In a phase ½ randomized

4. The limitations of Antibody-based anti-VEGFA therapy were not well discussed, such as anti-vegf resistance.

Reviewer 2 Report

The review describes the clinical data on the treatment of AMD, presents the results of human studies of various techniques and drugs. Undoubtedly, these data will be of great interest to both clinicians and scientific researchers in this field. The review is written in an understandable language, well structured and complete.  

It is worth noting that the review does not provide links to two articles from which information was used: 

Chong V. Ranibizumab for the treatment of wet AMD: a summary of real-world studies //Eye. – 2016. – Т. 30. – â„–. 2. – С. 270-286.

Schmidt-Erfurth U. et al. Guidelines for the management of neovascular age-related macular degeneration by the European Society of Retina Specialists (EURETINA) //British Journal of Ophthalmology. – 2014. – Т. 98. – â„–. 9. – С. 1144-1167.

Request to authors to add references to the bibliography.
